# The Experience of Home Parenteral Therapy: A Thematic Analysis of Patient Interviews

**DOI:** 10.3390/pharmacy11050133

**Published:** 2023-08-22

**Authors:** Marko Puzovic, Hana Morrissey, Patrick A. Ball

**Affiliations:** 1School of Pharmacy, Faculty of Science and Engineering, University of Wolverhampton, Wulfruna Street, Wolverhampton WV1 1LY, UK; marko.puzovic@nhs.net (M.P.); patrick.ball@wlv.ac.uk (P.A.B.); 2School of Dentistry and Medical Sciences, Charles Sturt University, Wagga Wagga, NSW 2678, Australia

**Keywords:** Health Education Impact Questionnaire, home parenteral therapy, injectable medications, self-injecting, homecare

## Abstract

Background: A limited number of studies have explored patients’ experience with home parenteral (injectable) therapy (HPT) in the UK. Aim: To explore the immediate-, short-, and long-term experience of patients with self-management of any home parenteral therapy with the intention for developing a guideline for service development in the United Kingdom. Methods and design: An interview-based study of patients receiving HPT. Invitations were posted to all patients on the hospital HPT register. The sessions were conducted by telephone for all consenting patients. The interviews were recorded, transcribed, and analysed thematically. Participants completed the ‘Health Education Impact Questionnaire’ (heiQ) before and after the education session. Results: Of the 640 patients invited to participate in the study, 45 (7%) patients completed the interviews and the education session. An interview analysis revealed that the patients’ experiences of HPT were generally positive, but the levels of training and support received showed wide individual variations. The patients had experienced periods of doubt and uncertainty, where they would have appreciated quick access to professional advice to alleviate their concerns. There was a reliable positive change (10.5–18.4%) from before and after the education sessions in six out of the eight domains on the heiQ questionnaire (health-directed behaviour, self-monitoring and insight, constructive attitudes and approaches, skill and technique acquisition, social integration and support, and emotional distress) and moderate change in two domains (5.3% in positive and negative engagement in life, and 2.6% in health services navigation). Conclusion: Self-administered parenteral therapy at home is a valuable option, but training and preparation standards should be optimised across hospitals and the wider NHS.

## 1. Introduction

Increasingly, patients are being discharged from hospitals in the UK to continue a range of injectable therapies for a wide range of conditions, in their own homes. This includes subcutaneous thromboprophylaxis after surgery, antibiotics, ‘biologically derived medications’ such as monoclonal antibodies, cancer chemotherapy, and parenteral nutrition. Administration routes include infusion, intravenous, intramuscular, and subcutaneous administration. In theory, patients undertaking such therapy are trained, with their technique validated before or very shortly after discharge; anecdotal evidence suggested a wide variation in the preparation that patients were receiving, ranging from comprehensive to none at all. 

A limited number of studies have explored patients’ experience with home parenteral therapy (HPT) in the UK. One area of HPT studied in the UK is outpatient parenteral antimicrobial therapy (OPAT) services. Twiddy et al. [1] in Northern England reported that, even though OPAT provided opportunities for improved cost savings, the implementation was patchy, and a variety of service models were in use in the UK. This study looked at the experiences of patients receiving only one specialised form of HPT, and may not be representative of the whole population of patients receiving homecare parenteral therapy in the UK. Another UK study, by Thorneloe et al. [2], explored individuals’ perspectives of their psoriasis, medication, and its management. It involved qualitative interviews with 20 psoriasis patients; while only one patient was self-injecting a ‘biologic’ therapy, this is a growing area of HPT in the UK. 

Another expanding area is rheumatology. Chilton and Collett [3] used a mixed-method research methodology including a postal questionnaire with 109 patients, and one-to-one interviews with seven patients managed by a National Health Service (NHS) rheumatology department, to explore rheumatoid arthritis (RA) patient treatment preferences when faced with three anti-TNF-α parenteral therapy options. This research was focused on finding whether RA patients wished to participate in decisions about choosing their therapy, and the factors that influenced their choice, rather than on patient experience with the self-administration of HPT and home care. The study concluded that, if a larger number of patients were to receive treatment at home, rheumatology services would need to provide more patient education and support in decision making. 

Currently, little is known about patients’ experiences and perceptions about their training and education on self-administering HPT in the UK [4]. Originally exclusively in hospital, training is increasingly being outsourced to commercial clinical homecare providers, and HCPs who initiated patients on the long-term self-administration of HPT might not be directly involved in patients’ training and supervision, or the assessment of competence for self-injecting. In the USA, Potera [5] found that 84% of patients receiving therapy with autoinjector devices at home used them incorrectly, while more than half of those who made errors missed three or more steps during the process of self-administration. In addition, the ‘forgetting curve’ theory suggests retention and recall of information worsen over time without practice and repetition, which could mean that 50% of the information that HCPs give to their patients during any initial training about self-injecting might be forgotten within one hour, 80% might be forgotten in two days, and 90% might be forgotten in a week [6].

It was, therefore, deemed appropriate to explore the experience of patients on any self-injected home therapies to inform the gap in knowledge in this area. This research explored patients’ and carers’ experiences with HPT and aimed to include patients on different types of HPT, putting emphasis on their training and education, and the support received during the transition to HPT.

The study was approved by the UK Health Research Authority Ethics Committee, the Life Sciences Ethics Committee at the University of Wolverhampton, and the Local Trust from where patients were recruited. 

## 2. Aim

The aim of the study was to explore patients’ immediate-, short-, and long-term experience with self-management and HPT with the intention of developing a guideline for service development in the UK. 

## 3. Methods and Design

This was a qualitative, observational study of patients who were receiving any form of parenteral (injectable) therapy at home. The study interview and educational sessions were conducted over the telephone for all consenting patients. The study project steps are shown in Figure 1.

Currently, patients who become stable but require long-term injectable therapy are discharged to continue treatment as hospital outpatients where they attend clinics to receive their treatment, or as homecare outpatients where a nurse attends their home to administer the treatment (e.g., cancer patients, usually when the treatment is infusion or intravenous or intramuscular therapy), or as homecare patients who are trained to self-administer their own treatment (usually when the treatment is administered through a pre-established central or peripheral line where the patient simply connects the pre-prepared treatment into a port, either intramuscular or subcutaneous). The decision is made by the hospital treating team, and then the patient is informed of the outcome. The training is usually carried out by the ward nurse or a third-party medication supplier and can be either implied, e.g., the patient having seen it performed while they are in hospital or as a structured session upon discharge. However, while training is an essential requirement before discharge or the commencement of HPT, the actual content of what should be delivered or what the patient should become competent in is not standardised or written anywhere.

The guide for the patients’ semi-structured interviews was developed by the researchers. It was, then, reviewed by the Trust service users’ representative (twice) and amended accordingly [7], and the final version was pilot-tested [8]. As this was an exploratory and preliminary study, the semi-structured interview guide was not validated. The health education impact questionnaire (heiQ) was used to assess patients’ satisfaction with this study and the education session component [9]. The provider validated the heiQ tool and concluded that the eight independent dimensions have high construct validity and is a reliable measure of the benefits from a patient education program [9]. A total of 640 patients were posted a pack containing the invitation letter, the information sheet about the study, the consent form, the education discussion debriefing form, the baseline heiQ questionnaire, and the invitation to participate in the interview and educational session. Patients were asked to return the signed consent form in a return-addressed prepaid envelope if they wished to take part in the study. Patients who consented were then posted a second pack containing the draft educational booklet about HPT (Appendix A) and the relevant patient information leaflet about their particular parenteral medication, to read before the interview, and the heiQ questionnaire. These patients were contacted by telephone between April 2020 and October 2021. The participants were reminded they could request a break during the interview, or the interview could be postponed or cancelled if they felt it was stressful to discuss their medications or medical condition, or if they felt emotionally uncomfortable. Patient consent for audio-recording was verbally confirmed at the beginning of each interview. The interviews were recorded digitally using a Sony^®^ ICD-UX560 digital recorder (Tokyo, Japan) with an Olympus^®^ TP-8 telephone pick-up microphone (Hamburg, Germany). The digitally recorded interviews were downloaded and imported into Sony^®^ Sound Organiser (Version 2.0.03) computer software. Audio recordings were transcribed verbatim and anonymised. In order to facilitate the process of transcription, a Sony^®^ FS-85USB Foot Pedal/Foot Control Unit was used. All transcripts were double-checked for accuracy and to ensure patient confidentiality was maintained and all material removed from the transcript that could possibly identify individuals. Non-verbal and emotional elements of the conversations (such as laughter, long pause, confusion, etc.) were not strictly included in the transcriptions, because the study was mainly focused on the content of what the patients said, rather than making interpretations of the way this was said [10]. Transcribed audio-recordings were uploaded into a computerised assisted data management program (NVivo^®^ Pro software, Release 1.6.1, QSR International) to assist with creating codes, organising and summarising data, searching for interrelationships between codes, and identifying themes. Interview transcripts were converted to ‘cases’ and coded in NVivo^®^, which then allowed us to perform the qualitative data analysis. During the content analysis of the interviews, the transcripts were read and re-read several times (line-by-line), while the significant text segments were coded, i.e., flagged with a short phrase that symbolically assigns a summative attribute for that portion of the data. “Coding provides a means of purposefully managing, locating, identifying, sifting, sorting, and querying data.” [10] The codes were then categorised into main themes, and similar subthemes clustered into the related main themes. Relevant interview quotations are shown with an ellipsis (...) used to indicate omitted words or phrases, and brackets [ ] used to note words added for clarification.

There are a few broad approaches to qualitative analysis: thematic analysis (TA), grounded theory (GT), interpretive phenomenological analysis (IPA), and framework analysis (FA) [11]. In order to identify key themes relating to patients’ experiences with HPT, patient interviews were explored using an inductive TA approach. TA is widely used in health care research, where the recurrent patterns (themes) in the data content are identified (labelled or ‘coded’), analysed, compared, and categorised [12,13]. 

As the purpose of this research was primarily exploratory, the reflexive (where the researcher reflects on their assumptions) TA approach was deemed to be sufficient in order to describe the observed thematic groups, i.e., key issues of concern to a particular group of people [11,14]. 

## 4. Qualitative Analysis of Patients’ Interviews

Overall, 640 patients were invited to participate in the study, 54 (8.4%, mean age in years 54.2 +/−14.9, range 22–85) patients agreed to a telephone interview and educational and feedback session, but only 45 (7%) patients gave their consent and completed the interview. Of these, 9 patients could not be contacted for the interview due to various reasons such as the COVID-19 pandemic and becoming seriously ill. The telephone interviews took, on average, 13:10 min/s each (minimum 05:00 min; maximum 37:51 min). The total duration of all interviews was 9 h and 53 min. The interviewed participants and their demographics are shown in Table 1.

During the interview, patients were asked several questions to elicit their view on the quality and usefulness of the educational material, its content, their opinion about translating the material into multiple languages, and the benefit of creating a free ‘App’ or Internet page, as well as their experiences related to the training and support given when they first started their injectable treatment at home.

### 4.1. Was the Educational Leaflet Informative?

Out of 45 patients, 42 patients (n = 42, 93.3%) answered that the educational booklet and patient information leaflet were informative. The participants said that “*it explains pretty much everything that you would want to ask*” (P34, 45 years, male, MS), “*it’s very comprehensive (…) really well-structured*” (P183, 59 years, female, psoriasis), “*very informative*” (P410, 68 years, female, high cholesterol), and “*well-written (…) for somebody who is not a medical person (…)*” (P413, 58 years, female, high cholesterol). The content was described as “*very useful*”*,* especially the part “*about side effects*” and “*the bit on red flags*” (P535, 67 years, female, IBD). P445 (71 years, female, high cholesterol) said that the leaflet “*goes into more detail than I have had in any information before*”. One of the patients, who was experiencing pain during the injection at the injection site, explained that the advice from the booklet helped her to reduce the pain:


*… um... but **this leaflet that I’ve got here from you, I found some useful tips**... Um, talking about page 16, ‘one of the most common side effects of injections under the skin is pain at the injection site’… um, and I’ve found that that is the case when I am injecting, not afterwards particularly but when I am actually injecting. **So, this advice to allow the medication to warm to room temperature for 15–30 min was helpful information and I have tried that, and it has helped**… um, and so on. So, yeah, I have revisited that to make sure that I’m doing everything… to make it as least painful as possible.*

*(P622, 63 years, female, high cholesterol)*


It is not clear whether the advice was not given to the patient during the initial training, or she had forgotten about the advice given. This clearly shows that regular assessments of patients’ technique should be part of the follow-up process in patients on long-term injectable therapy at home. Other patients commented that reading the leaflet was a good reminder about their injectable therapy because people just “*forget things*” with time (n = 3):


*… but, you are kind of, just get used to things and forget the details, so it was helpful to read up on that again. *

*(P313, 22 years, female, IBD)*


Having a lot of information available can also be overwhelming for some patients, and they might be struggling with interpreting and understanding the information given, or they might not be interested in reading the information provided (n = 2):


*It was. Uh, it was **quite, um, a lot of information to take in**. Um, but I did feel like it did give me everything that I needed to know. *

*(P59, 33 years, female, MS)*


However, patients (n = 3) explained that having a booklet they could read and re-read at their own pace and in their own time is helpful for understanding better their condition, medication, and treatment:


*Yeah, it was very informative. It broke it down into little bits, it’s nice to have something written down. When someone tells you something you put it to the back of your mind or just take little bits from it but to have it written down in front of you, **you can read it at your leisure** and it’s quite handy. *

*(P630, 42 years, male, IBD)*


Some patients found the information about homecare service useful (n = 1), while another patient commented that the booklet and leaflet were informative but did not provide any new information that has not already been communicated during the training. The question remains whether HPT training varies between different patient groups based on their disease, or between different HCP teams providing the training. Only three patients responded that they did not learn any new information. One patient also mentioned that the content might be a bit complicated for some patients and they just might choose to disregard it:


*I feel like this small book… I think, it’s from my perspective, I mean, I understand it, but I think for lots of members of the public that go on this for the first time **it might be a bit complicated** for them, **or they might just disregard it** (…) when you first open it with all the indexes and things, um, it’s…. um, it could be perceived as, um… a booklet that you get sometimes for different things but **you think you know what you’re doing so you don’t always read through it**… It’s a bit awkward to… it’s like, um I mean I’ve read through it thoroughly and **I didn’t learn anything new from this booklet, but for some patients they might think it’s… that they don’t need to know any of that.**
*

*(P1, 59 years, female, IBD)*


### 4.2. Have You Got Any Questions about Any Parts of the Educational Booklet? Would You like Me to Go through All Sections or a Specific Section?

In general, patients did not have any specific questions regarding the educational material and were not interested in going through the sections of the booklet and leaflets. They felt “*quite confident that [they] have sufficient information*” (P34, 45 years, male, MS), that the booklet was “*very good (…) very basic language and very understandable*” (P160, 75 years, male, psoriasis), “*very clear*” (P183, 59 years, female, psoriasis) and “*quite self-explanatory*” (P204, 71 years, male, IBD and P260, 39 years, male, IBD), and “*so informative (…) that it explains everything that [they] needed to know*” (P445, 71 years, female, high cholesterol).

### 4.3. What Content Do You Think Should Be Added or Expanded in the Educational Leaflet?

The most mentioned topic (n = 4) was adding a section on when and how to get help. One of the patients suggested attaching a contacts sticker to the booklet with details on whom to contact and when to do so if any concerns arise during injectable treatment at home. It is evident that patients want clear and specific instructions on when and how they should contact their HCPs for help.


*Um, it might be interesting to know if I had any queries or if something went wrong, what was I supposed to do... (…) … Yeah. I would say, if something does go wrong, if you do make a mistake… you know, it would be nice to know who you should phone under those circumstances. *

*(P471, 72 years, female, high cholesterol)*


Another patient suggested adding the personal experiences of other patients on the same injectable therapy to the booklet. However, she acknowledged this type of information might be more suitable for an Internet page.

Information on what to do if a dose is missed was another suggested topic that a patient thought might be useful if added to the booklet. Another suggestion was to make the booklet available in electronic format so the patients could carry the information/booklet with them and easily access the information. One patient (P529, 63 years, male, psoriasis) mentioned that it would be helpful to add more information about having vaccinations during the treatment with biologic medications which can dampen the immune system. He has read the “Can I have vaccinations whilst on ixekizumab?” information in the supplied British Association of Dermatologists ixekizumab leaflet [15], and he was not sure what the difference between a ‘live’, ‘attenuated’, and ‘inactivated’ vaccine was, and whether the attenuated vaccine would put him at risk. The meaning of ‘attenuated’ was explained to the patient and he was reassured that he would not receive a ‘live’, also called ‘live attenuated’, flu vaccine but the ‘inactivated’ version, which does not contain any live flu viruses that could cause harm to someone who is on immunosuppressive therapy. 

Another practical suggestion, which is not related to the content of the booklet, was to increase the print size for people with sight difficulties or have it printed in Braille.

### 4.4. If We Translate This Leaflet into Multiple Languages in Addition to English, Would You See Any Additional Benefit to the Patients?


*The great majority of participants supported the idea of translating the booklet and leaflet into multiple languages (n = 41, 91.1%), and they thought this would provide additional benefit to non-English speaking patients. A couple of patients asked if the leaflet was also available in Welsh because they live in Wales and “cannot speak English”.*

*(P509, 39 years, female, IBD)*


### 4.5. If We Make a Free Educational Telephone App or an Internet Page, Would You See Any Additional Benefit to the Patients?

All patients said that they would see additional benefits to themselves and other patients. A few patients added that this might not be suitable for everybody, especially for older people or for patients who are not so computer-literate (n = 3) and they might prefer to ring the prescriber instead. There was an opinion shared by some patients (n = 6) that the Internet site and telephone ‘app’ would be read and used more in comparison to the paper booklet, while the perceived benefits were more up-to-date information, saving printing cost, easier to use, immediate information availability, and ability to reach more people, especially young people. Two patients said that people already use apps on their smart telephones for other things in their everyday life, and the telephones are always available:


*Yes, I quite Internet pages or phone apps because they are **easy to use**, aren’t they? **Because I would lose it because I just lost the booklet**, so, it would be fine. *

*(P385, 57 years, female, MS)*


One patient indicated that an NHS-certified app would give more credibility and reassurance to the patients of the quality and trustworthiness of the information provided. Five patients felt that the app “makes it kind of more of a personal medication experience and people [would] feel more comfortable knowing that they have got support just on the end of a telephone or (…) on a laptop (P313, 22 years, female, IBD). One patient’s view was that it is “*negligent for dermatology to place someone on pharmaceutical medication which treats the symptom and not the cause*” *(P162, 32 years, female, psoriasis).* This patient also shared her vision of what should be ‘at home medication’:


*I do think that it is an aspect of responsibility that should be written about, about lifestyle and nutrition within these types of leaflets if they are sort of “at home medication”. ‘Home’ is a place whereby the understanding of stress relief, rest, and good nutrition should be taken into account far more seriously than the reliance of injecting something into your body. *

*(P162, 32 years, female, psoriasis)*


### 4.6. What Do You Think Would Be the Most Helpful Content for You and Other Patients within the App?

The thematic framework summary for this question is shown in Table 2.

Patients (n = 41) wanted to be able to contact the HCPs for some advice online, because they feel they could express themselves better through typing questions on a computer device:


*…perhaps some **people don’t open up or talk as much as they would** like to and some say ‘yeah, yeah, yeah, that’s fine’ and then they finish the conversation and they think **‘oh, I forgot to ask about that or add that’** so if there was possibly a webpage or something with a little bit of extra information or again you know a different contact number then you know… that’s why when you go to the Doctor’s they recommend that you write notes isn’t it, because you know, **you can lose yourself in the moment and forgot the crux point that you actually needed to ask**. So yeah, I think, um, a website or a contactable thing would be handy.*

*(P80, 54 years, male, MS)*


A few patients mentioned a ‘forum’ or ‘live chat’ would be beneficial, as they could discuss their issues with other patients who are injecting the same therapy. They find it easier to communicate with other patients rather than with the HCPs, as they feel that discussing their problems with people who are going through the same experience would provide encouragement, offer troubleshooting ideas, and make them feel more supported and less alone. 

However, some patients would prefer to be able to contact the HCP online when they need support, especially if they could just type a question and receive a quick response over the Internet, like in the following case, where the patient could not administer the injection because of a previous painful experience. Patients would feel more supported if they were given the option to contact the HCPs if they experience any issues while being on injectable therapy at home:


*For me, personally, just **details of who to contact if I have a problem with it (…) if something does go wrong, if you do make a mistake**… you know, it would be nice to know who you should phone under those circumstances.*

*(P471, 72 years, female, high cholesterol)*


The second most discussed topic of importance for the telephone app or Internet site was to have a section on side effects, which they can access especially when starting treatment, when they might feel more scared and concerned as they cannot tell the difference between a less severe and more severe side effect, or how to recognise a side effect:


*I think that the most worrying thing is, you know, when people are actually administrating the drug and what that is going to do… and I think that secondly, I mean people, from what I would think… are more **concerned about side effects**. So I think that things like **side effects should be made really clear so that people understand what is happening**… because I am probably a bit more fortunate but some people are not so they may be a bit more concerned about side effects and what is going on… um, maybe if they could look at that on the Internet very quickly, perhaps it would give them some solace.*

*(P157, 74 years, male, psoriasis)*


Alongside side effects, patients would like to have an understanding on how the drug works so they could gain a better understanding of *‘what they are putting in their bodies*’. In addition, they would like to be able to read some advice on what to do if they experience a side effect, when to report it, and when to seek help and from whom. The availability of a video demonstrating the administration technique, i.e., how to self-administer the injections, was listed as helpful when patients are starting injectable therapy and would also provide guidance on which sites on the body can be used for injecting. 


*(…) if their helpline is busy you can always then refer to the Internet and perhaps see, they could perhaps put on the Internet (…) **somebody showing how the pen works** (…) … just very short video and (…) the **areas of the body where it could go into** and (…) **how quick it is to use** it, that type of thing (…) because I don’t think I had been pinching the skin enough on my thigh when I was doing it initially… and that’s why I got the raised bump. *

*(P413, 58 years, female, high cholesterol)*


Apart from the injection technique videos, some patients would like to see some brief videos which would explain how their drug works in the body. Three patients discussed having a frequently asked questions (‘FAQ’) section on the website, as well as building a knowledge database from previously asked questions.

Another helpful feature, suggested by two patients, would be to have a reminder on the telephone app for when to take their injection, or when to order new supplies from the homecare company:


*I’m just thinking about making it a personal app, maybe some **ability to store information** about… because of the four weekly cycle… this might be daft as people have calendars and things but… I feel a reminder service… A reminder system so that it’s ‘make sure you have ordered’… actually the company is pretty good at getting in contact with me but also **a reminder to take it**.*

*(P529, 63 years, male, psoriasis)*


### 4.7. When You First Started Your Treatment, What Do You Think Was Missing? How Could Healthcare Professionals Have Done Better to Improve Your Confidence and Competence during Training?

The thematic analysis for these two questions is summarised in Table 3.

### 4.8. Challenges with HPT

When asked about what was missing and how could HCPs have done better to improve their confidence and competence during the training, the majority of patients (n = 33, 73.3%, mean age in years 56.4 +/−14.4, range 23–78, coded 47 times across 33 interviews) used words with positive meanings about their experience with their initial training of injecting at home and the homecare service. They said that the “initial setup was excellent” (P1, 59 years, female, IBD), they “had sufficient support and information” (P34, 45 years, male, MS), and were “very happy with what (they were) told to do”, and the healthcare professionals were “absolutely 100% first class (…) and (…) absolutely brilliant” (P67, 56 years, female, MS). Some patients said they “believe that [healthcare professionals] covered everything at the time” (P80, 54 years, male, MS).

P188 (52 years, male, psoriasis) described his view of the benefits of the homecare program as “a personal approach to patients and a well-organised logistics that provides a sense of being well-supported by the homecare service which eliminates the need to attend hospital appointments for treatment.”

Patients explained that “if you do have a problem, you can telephone up them [at the hospital] and they can give you advice or (…) calm you down or (…) give you reassurance” (P160, 75 years, male, psoriasis). Some homecare nurses provided support and reassurance by telling patients after training that they “could ring [them] back and [they] would come back out and do it again [with them]”, and the patient felt it was “nice to have that as a backup” (P253, 35 years, female IBD). P410 (68 years, female, high cholesterol) commented that her training was a “very good experience because someone actually came and spent time with me explaining all about the injection, how it worked, and how to administer it… … myself”. P188 (62 years, male, MS) commented that he was “quite overwhelmed” when this method of treatment was suggested to him, but, “after the training was carried out on the first injection, [he] was surprised how competent and confident [he] was with ease to do it”. Patients also reported that homecare treatment was “allowing independence and freedom” (n = 4), as “you can do it in your own time, you do not have to travel to a hospital” (P121, 46 years, female, IBD), or to their “Doctor’s Surgery” (P183, 59 years, female, psoriasis), along with “the fact that [they] could do it at home” (P596, 48 years, female, IBD). 

Despite all the positive experiences and perceived strengths of the homecare treatment, as Table 4 demonstrates, patients also faced several challenges at various stages during HPT.

The most discussed sub-theme was concerns about side effects. Some patients did not want to bother their HCPs with side effects and stated that “*it is quite difficult to determine whether the side effect is severe enough to actually get in touch with the doctor or whether you should just wait and see*” (P535, 67 years, female, IBD). They were also concerned about how to respond to the side effects of their injectable therapy when they are at home and alone. Some patients felt that they had not been adequately informed or supported about side effects by their HCPs:


*I think, also, one of my worries when I take medication myself is that is **allergic reactions and how to respond to that when you are at home**… sometimes I’m on my own, at home, and so there is no one with me and I have, in the past, had an allergic reaction with Infliximab in the clinic… but **I did worry when I started to take Stelara jabs at home, what would happen and what to do if I had an allergic reaction on my own, at home…***

*(P627, 48 years, female, IBD)*


Patients who were injecting immunosuppressive therapy (e.g., for psoriasis or IBD) were concerned about side effects from such therapy. They were aware that the drug can lower their immune system and were worried about the impact of this during the COVID-19 pandemic, or whether this immunosuppression could cause or might have caused cancer:


*… I mean really, I wasn’t aware, with the Covid thing, what an impact the side effects of it being an immune suppressant would be in this situation and that’s, you know, the fact that I have had to shield and continue to shield for such a long time wasn’t something I had thought about at the beginning of taking it. *

*(P167, 64 years, female, psoriasis)*


As the interviews were conducted during the pandemic, eight patients described various challenges that they were facing whilst being on HPT. Patients reported their concern about being on immunosuppressive biologic therapy and the implications of catching COVID-19:


*And another thing I want to say is, well, when obviously… this probably will never ever happen again, but when there is a pandemic like this, that affects so many people, it would be good to get a phone call to give people more information. I’m on the 12-week NHS list anyway, but I didn’t get that letter until four weeks into lockdown, but it would be nice if somebody phoned, or the Teams phoned people that were on the biological, because I didn’t know whether to stop it, or whether to continue with it. Obviously, I stopped it when I had COVID-19 ‘cause I was too ill, but I don’t know how… I would have liked to have known more information about how COVID-19 was going to affect me with the weakened immunity… *

*(P1, 59 years, female, IBD)*


The second most discussed sub-theme was injection site problems such as injecting technique, pain, or injection site reactions like bruising, swelling, or rash when self-injecting. These problems might be stemming from inadequate initial training and the lack of patients’ competency assessment by the HCPs. P204 (71 years, male, IBD) reported confusion because he had received different advice about the injection technique from HCPs on whether to pinch or stretch the skin during the injection; he decided to go back to the original instruction to pinch the skin as stretching was too painful, but he was not sure if this change would have a negative effect on his treatment:


*Uh, well at the hospital they told me to pinch the skin and inject into the thigh or the belly. And they told me to pinch the skin up and then to inject into that. Whereas, when the district nurse came out, she told me to actually stretch the skin between the thumb and the finger, stretch it and then inject into it there… two completely different things… (…) that was a little bit baffling as to why they had changed it.*

*(P204, 71 years, male, IBD)*


One patient had concerns about injecting into the ‘belly’ as he viewed it as being too invasive and more painful compared to the thigh. The doctor advised him again to try injecting into the belly, which improved his experience and reduced injection site problems.

Several patients commented that pain, stinging, and bruising got worse when they were switched from the patented brand to a ‘biosimilar’ preparation, namely, from the originator brand Humira^®^ to Hyrimoz^®^ adalimumab biosimilar (Chicago, IL, USA). The HCPs’ explanation to patients was that the biosimilar injections (Hyrimoz^®^) had a larger needle. However, the increased incidence of injection site reactions, like pain and stinging, could have been attributable to a combination of factors like the presence of citrate in the Hyrimoz^®^ formulation which was removed from the updated Humira^®^ formulation, a larger injection volume (40 mg/0.8 mL Hyrimoz^®^ vs. 40 mg/0.4 mL Humira^®^), a larger needle size (27 G Hyrimoz^®^ vs. 29 G Humira^®^), and, potentially, a difference in the administration device (Hyrimoz^®^ pre-filled syringe vs. Humira auto-injector) [16]. 

Additionally, the speed of injection, fluid viscosity, injection angle and technique, product temperature, allergens, injection frequency and injection site, low body weight, injection anxiety and needle phobia, pain catastrophising, nocebo effect, fibromyalgia, and depression may contribute to HPT self-injecting problems [17]. Some patients were not aware “*that you’re supposed to take it out of the fridge 15 min before to reach room temperature before you inject*” (P509, 39 years, female, IBD). Patients also reported conflicting advice they received from HCPs about the use of alcohol wipes, where some patients were taught to swab the injection site, but others were told that this is completely unnecessary. The same patient commented that she used to get the alcohol wipes with Humira^®^ adalimumab, but not with the Amgevita^®^ or Hyrimoz^®^ biosimilar, and that she is now considering buying them online. When this issue was discussed with the hospital, they confirmed that the supply of alcohol wipes was part of the Service Level Agreement (SLA) between the hospital and the homecare company that was supplying the injections to the patient, which had been overlooked by the supplier of the biosimilar product. The company replied with an official apology and promised immediate corrective action. 

There is still an ongoing debate regarding skin preparation and the use of alcohol swabs to clean injection sites [18,19]. The World Health Organization [20] suggested the use of 60–70% isopropyl alcohol or ethanol, but that it was not necessary if the skin is visibly clean, while other authors recommended using alcohol swabs to prepare the skin in older patients and those who are immunocompromised [21,22]. 

Problems with the injection device were reported by three participants such as bent needles, auto-injectors not ‘firing’, or delivering an incomplete drug dose, as well as faulty batches of pens. 

Two patients compared the auto-injectors with pre-filled pens and trigger injectors. It is very important to select the most appropriate device for an individual patient in order to improve adherence to therapy as well as to minimise any usability problems or use-errors [23]:


*I am using the auto-inject… and I do find that is far far better and far easier to use [compared to the] pre-filled syringes *

*(P160, 75 years, male, psoriasis)*


The third most discussed sub-theme was anxiety with self-injecting and fear of injections, which is remarkably common among the population but sometimes underestimated by HCPs. Some studies have shown that prevalence estimates for needle fear ranged from 20–50% in adolescents and 20–30% in young adults and patients with chronic conditions requiring injection [24]:


*It’s a long time ago. I mean, I remember doing **the very first injection which I have to do in the hospital… the self-injection and I actually fainted almost immediately afterwards**… um, but that was probably more of the shock than anything else and having done it. I mean, I’m not a great fan of injections… *

*(P92, 50 years, male, MS)*


One patient commented that “doctors just think ‘oh send them home and they will inject themselves’ because to them it is an everyday occurrence, but, to a patient, to inject yourself is (…) traumatic” (P157, 74 years, psoriasis):


*Then the second time I did myself, and **I am absolutely terrified of needles, the point of people have got to pin me down**. Oh… and I’ve… yes, they were absolutely brilliant. That is a major, major step and it’s **all because of those healthcare professionals at the beginning.**
*

*(P67, 56 years, female, MS)*


Even with the support from HCPs, some patients reported that they were not able to self-inject for quite some time after starting their therapy. They required someone else to administer their injections, for example, a homecare nurse or family member:


**
*Initially, I couldn’t do it to myself, you know, I was like ‘ooh no, I can’t do this to myself’… and my husband used to do it*
**
*… but now, it’s like ‘just do it’ you know?! … **I had had a long-standing thing about vaccinations and needles and things from childhood**… you know, um, I’m wonderful now... I’ll have a blood test and it doesn’t bother me now, … *

*(P167, 64 years, female, psoriasis)*


Six patients reported confidence issues with self-injecting after being supervised for the first injection only. One of these patients had her initial training session delivered by a nurse through a video link, and just once. She said that “*it would be really worrying for some people [to inject themselves after only one session* via *video link… it’s just that they’re not confident to do it*” (P609, 73 years, female, high cholesterol). Six patients have expressed concerns about the efficacy of their treatment as their disease got worse after changing from the patented drug to a biosimilar injection for IBD. Four patients expressed their skepticism towards treatment options. For example, P535 (67 years, female, IBD) commented that she is “*probably I am at a stage in my life where I do not automatically accept that every medication that is suggested is good for me. I question it. Whereas when I first started with the Crohn’s I probably would have accepted anything without any questions.*” P529 (63 years, male, psoriasis) said that he was “*offered the possibility of injections many years ago but at [that] time*” he “*was not ready*”; now, as he has “*gone through nearly everything, that is available… to treat psoriasis he has no other option*”*:*


*Funnily enough, **since they have changed it, it doesn’t seem to be as good as before** and in fact, when I first started on the brand x, I was injecting once a fortnight but now I have gone to every week, **I have to inject every week now**. Um, when I was doing the brand x, it was virtually normal. It was great. I had no bowel problems whatsoever but it’s not quite as good and I just thought it was basically to do with Crohn’s getting worse.*

*(P204, 71 years, male, IBD)*


Another challenge reported by four patients was dealing with mistakes or troubleshooting while being on HPT. One lady got distracted and injected herself into her thumb and then hesitated to contact the HCPs for advice. P627 (48 years, female, IBD) shared the same concerns and commented “*… what would happen and what [would I have] to do if I had an allergic reaction on my own, at home…*”. P260 (39 years, male, IBD) explained that one of his concerns was what to do if he ended up missing a dose, which was explained to him during his initial training, but he could not remember. Another patient on HPN described his confusion on what to do when he first experienced a fever related to his infected central line. 


*… **if you get a temperature**, if you get a fever… this is what you need to do (...) but the first time that this happens, **it is ……… scary** to say the least. Um… **and you don’t really know what to do**. So, to have a kind of step-by-step ‘this is what you do’, ‘take yourself off your PN’, ‘call your nurse’, ‘if it’s out of hours you need get to hospital’…*

*(P638, 35 years, male, home PN)*


### 4.9. Communication Issues

Patients (n = 15) complained about their inability to contact HCPs for advice “because, you can sometimes feel brushed aside at times and it [would be] nice to know that there is just a contact number that you can ring up or email and say ‘look, I’m not sure about this, can you please clarify?’” (P80, 54 years, male, MS). A timely response from the HCPs is also something that the patients would like to receive while being on HPT as “you can’t always get a telephone call straightaway” (P567, 31 years, female, IBD), and “it’s always (…) a 12–24 h delay before [they] can get an answer” (P92, 50 years, male, MS). P473 (68 years, female, high cholesterol) suggested that, if the patients “were able to just type a quick question to somebody and they came back fairly quickly, I think that that would be useful rather than trying to make a telephone call or trying to find somebody to speak to”:


*I suppose the only thing that would be (…) useful would be **if there was somebody either at the end of the telephone or face-to-face to (…) review what training has been done and for that person to test or check that it’s being done correctly.** (…) The training I have had was literally a training pack… with instructions in it and a sample of the injection… and I had no contact with anybody. So, I just had to do it myself.*

*(P465, 68 years, male, high cholesterol)*


P529 (63 years, male, psoriasis) expressed his concern that he was never seen by a consultant when starting his HPT; he only met the nurse who is working for the consultant. Another patient (P471, 72 years, female, high cholesterol) commented that she was never told what she was supposed to do if she had a query or a problem. On the other hand, eight patients have expressed their *reluctance to reach out for help*, because they do not want to “*really bother the medical professionals*” with their problems (P67, 56 years, female MS) and “*because you are not the only patient on their list*” (P535, 67 years, female, IBD). P439 (49 years, female, MS) felt that she would not want to bother the HCPs with her problems unless she had been given clear instructions on when and how to contact the HCPs, because “*[that] almost gives you permission that it’s OK to ask*”. 

Feeling isolated was another challenge reported by six patients. P1 (59 years, female, IBD) felt quite isolated with her IBD as she “*[did not] know anyone else that’s got it*”, while P67 (56 years, female, MS) commented that “*it’s a nice feeling that you are not on your own*”. P80 (54 years, male MS) explained that “*a contact number that you can ring up or email (…) helps personal wellbeing as well*”. Patients also reported receiving contradictory messages from HCPs. 

P509 (*39 years, female, IBD*) raised her concerns about why her GP and the hospital cannot access each other’s information (like blood tests) even with her permission, just because she and her GP live in a different county in England to the hospital. Another patient commented that her treatment was delayed by eight weeks because there was a communication problem between the homecare company and her hospital team (*P627, 48 years, female, IBD*). P630 (*42 years, male, IBD*) reported that his training and sign-off for self-injecting only happened over the telephone.

The perceived lack of support, guidance, reassurance, and follow-ups was discussed by 22 patients. P1 (*59 years, female, IBD*) said that she only received the training for the first device but not the second or third brands of devices:


*Possibly I would have understood that there would have been more support available if I needed it but I didn’t ask for it at the time, so the whole first series of injections, because I was on daily injections at the time, um, yes, **there was actually no support from anybody apart from the first day**… … so it might have been nicer if (…) **somebody could have got in touch with the first couple of injections** or just on the day that I should have done an injection **just to check with me that I had done it and that I was still comfortable with it… I was but I was never given the chance to say that I wasn’t**, proactively, if you know what I mean? I’m sure I could have rung somebody and told them it was not working for me but, yeah **it would have been nice to have a catch-up call…***

*(P92, 50 years, male, MS)*


One patient described how she struggled with self-injection in the beginning, and that she would have preferred to have some kind of support or supervision to assess her technique. A more standardised and structured training and counselling would also improve patient experience and improve patients’ self-assurance and confidence:


*… I mean when you read about it, the side effects that you can get from it, and it is… as a person, you look at that thinking **‘well why should I be taking something that seems to be quite dangerous!’. And I mean you do need quite a lot of reassurance on that**.*

*(P535, 67 years, female, IBD)*


Patient P636 (85 years, male, HPN) reported that he stayed on a 24 h infusion for 5 years. The nurses came to disconnect his HPN bag and then immediately connect the next one, which was convenient for them but not for the patient, and he did not receive a single follow-up by the consultant. He was bed-ridden and could not leave his house for 5 years until another (local) hospital took over his care as he put on too much weight. They immediately reduced his HPN infusion time to 12 h and provided nursing support twice daily. 

Twenty patients have also described challenges with access to information while being on HPT. They reported struggling to understand the terminology, being overwhelmed with the given information, struggling to find reliable and trustworthy information and sources of information online and being worried about reading some information about side effects:


*I think that… as I say, **when I initially got told I was going to be injecting I think I thought of the wrong thing completely, so I think when I was first told, maybe then I should have been explained it differently or given some information** there and then to know, actually it’s going be like an epi-pen that’s going to be a ‘click’. That it’s going to be relatively easy (…) When you’re not a trained (…) nurse yourself, it’s quite scary.*

*(P253, 35 years, female, IBD)*


### 4.10. Problems with Travelling and Special Storage Requirements (Refrigerator)

Biologicals are potent protein-based drugs and there are strict rules regarding their transportation and storage. Extreme temperatures and vigorous shaking should be avoided. These drugs should be stored refrigerated between 2 and 8 °C because extreme temperatures can lead to protein denaturation and irreversible protein aggregate formation. Protein aggregates are known to induce the body’s immune response to the therapeutic proteins, with the formation of antidrug antibodies, which may reduce the clinical effectiveness and/or result in the treatment failure of biologicals, or even cause hypersensitivity adverse drug reactions (e.g., anaphylaxis) [25].

Patients (n = 15) reported challenges with travelling and special storage requirements (refrigeration between 2 and 8 °C). P92 (*50 years, male, MS*) commented that “*… we have been on holidays and camping, which I was always nervous about [because of the cold storage and transport requirements]*”. Some patients even commented that they would not know “*what to do when you go on holiday about the storage of the drug*” (*P240, 39 years, male, IBD*). Another patient has described her unpleasant experience with travelling abroad with injections in her luggage and going through security check (*P44, 60 years, female, MS*). Some patients might be avoiding travelling abroad due to the challenges and inconvenience with refrigerated transport and storage:

***I’ve never looked at having to take it away abroad or anything like that.***  
*And to be honest, probably if I was going to do that, I would probably time any holiday around the four weekly cycle rather than have to take it away and put it under…*
*(P529, 63 years, male, psoriasis)*


The refrigerated delivery and storage of injections is an issue for patients who are working and are not at home when the homecare company delivers their therapy (*P529, 31 years, male, IBD*).

Patient P260 (39 years, male, IBD) was worried about the efficacy of his medications as he left the syringe out of the refrigerator for 45 min at room temperature before administration, instead of 30 min as told by the HCPs. He was given reassurance during the interview that there would be no implications, as his drug can be stored at temperatures up to a maximum of 25 °C for a period of up to 21 days. However, it was also highlighted that this injection must be used within this period or thrown away, even if it has been put back in the fridge. Another patient asked if her practice of removing the injection from the fridge the night before administration and leaving it to warm up overnight is appropriate, because “*that is kind of like the same as leaving it 14 days or less [at room temperature]*” (P313, 22 years, female, IBD). 

The interviews revealed that patients have been instructed to store their injections in a domestic refrigerator. However, they have not been trained or instructed to monitor the refrigerator temperature or advised about the most appropriate location in the fridge in which to store their injectable drugs (e.g., to avoid accidental freezing). 


*The only question I did have when I was reading through… it did appear to indicate that **it would be better to have a separate fridge to keep the medication in rather than keeping it with the food** and that is not possible in my household. It is kept in the same fridge as all the other food… **that was never discussed with me before… it was the first time I had read about it on your leaflet today**. Yes, we do have a thermometer in it. But obviously it’s opened up frequently during the day because we are accessing food from it.*

*(P535, 67 years, female, IBD)*


A study from the Netherlands [26] found that, during transport and subsequent storage at home, only one in eight biological injection devices (11.6%) had been stored between 2 and 8 °C. In addition, 11.2% were stored more than 30 min below 0 °C and 33.2% were stored more than 1 week above 8 °C. Another study reported that only 6.7% of patients stored their biologic injection within the recommended temperature range, while 24.3% of patients stored their medication for more than 2 consecutive hours below 0 °C (median duration 3.7 h), and 2% stored them for more than 2 consecutive hours above 25 °C (median duration 11.8 h) [27]. 

P439 (*49 years, female, MS*) commented about the large size of the outer packaging of her injections as being problematic to store in a small domestic refrigerator, and some patients might be removing the outer packaging to fit their injections into the refrigerator. 

The Royal Pharmaceutical Society (RPS), National Clinical Homecare Association (NHCA), and National Homecare Medicines Committee (NHMC) have already reported that there is a lack of national guidance on maintaining the cold chain within patients’ homes [28]. 

A few patients (n = 4) described problems with deliveries and supply and dislike the idea of having their medications delivered to their home and having to wait for the delivery with a lack of communication. 


*Um, the only the thing that I found was missing was **the initial delivery of the medication** because **I wasn’t sure when that was arriving or when the nurse was coming** so that was a bit of… the injections did come first luckily and then the nurse came after. So, **it was a bit confusing**, should I say *

*(P614, 55 years, female, high cholesterol)*


## 5. The Health Education Impact Questionnaire (heiQ)

The heiQ was used in this study under the license agreement with Deakin University, Australia. The questionnaire covers two parts, baseline and follow-up, and consists of eight domains: health-directed activity, positive and active engagement in life, emotional distress, self-monitoring and insight, constructive attitudes and approaches, skill and technique acquisition, social integration and support, and health service navigation [29,30,31]. Following the completion of the baseline heiQ, which was sent to patients before the interviews, the follow-up questionnaire was sent after the interviews and education sessions took place. Of 45 interviewed patients, 38 (84.4%) returned both the heiQ baseline and follow-up. Completed questionnaires with their ID numbers were analysed and scored based on a Likert-type scale from 1 to 4, corresponding to ‘Strongly disagree’, ‘Disagree’, ‘Agree’, and ‘Strongly Agree’. Each domain had four to six questions. The scores were summed within each domain to obtain a scale score for each domain based on guidelines and the heiQ developer’s scoring protocol. In calculating the benchmarks, a small amount of missing data on individual heiQ questions were replaced with point estimates based on the questionnaire developer’s guidelines. The participants’ domain mean scores achieved in the baseline were compared with the follow-up mean scores to determine if patients’ knowledge improved after the telephone interview and patient education intervention, using the developers’ guideline and supplied Excel™ files. The demographics of participants who returned both the baseline and follow-up heiQ questionnaire are shown in Table 4. An overall analysis of the demographics of 38 patients who completed both questionnaires indicates that the majority of participants were women (n = 25, 65.8%), and the age range was 23–78 years (Mean = 56.4, SD = 14.4). The majority of participants (n = 24, 63.2%) were patients older than 55 years, self-injecting adalimumab (n = 15, 39.5%), and suffering from IBD (n = 12, 31.6%) or high cholesterol (n = 11, 28.9%).

Table 5 displays the proportions of program participants who achieved a meaningful change on eight heiQ scales according to the following benchmarks: (1) the baseline means; (2) the mean at follow-up; (3) the mean individual change score; and (4) an estimate of the proportion of participants who achieved a net positive reliable change. 

## 6. Discussion

The concept of HPT has many positive aspects for the patients as it can improve their care and quality of life whilst freeing hospital beds and reducing the healthcare cost [32]. However, Hackett’s report [32] also identified a range of clinical governance issues that might be associated with homecare as well as a lack of consistency in the homecare business. Based on this, the Royal Pharmaceutical Society (RPS) recommends activities that include patient training and competency assessment related to self-administration, requiring basic aseptic technique, special storage requirements, and training, all of which must be provided to patients and/or carers in the use of injectable therapy at home, and other support mechanisms must be in place, to ensure an effective use of homecare medicines and maximise treatment safety, effectiveness, and adherence [33].

However, the analysis of patient interviews in this study indicated that patients on HPT have varied experiences. Some patients reported that they received excellent support and training while others reported a lack of support, guidance, training, cold chain maintenance, and reassurance. This indicates that there is a lack of a standardised training approach between various homecare providers and the NHS teams. The maintenance of the cold chain within patients’ homes and its monitoring, inadequate training on injecting techniques, lack of competency assessment, and inability to contact the HCPs for help and advice are just a few of the issues the HPT patients experienced and revealed in their interviews. 

The participants raised some critical points and diverse opinions and reported a range of experiences with HPT. Patients with different underlying medical conditions or with a rare condition may have different opinions and experiences related to home injectable therapy. It was also evident that some basic information about the self-injecting technique is not always explained to the patients when they start their HPT. A lack of adequate competency assessment and training could affect adherence and lead to the improper use of the injection devices, incorrect dosing, and dose omissions, which can result in the increased cost of therapy and can potentially induce treatment failure. Patients who had received additional support from their HCPs during the initial training and supervision reported greater satisfaction with the homecare service. Therefore, patients should be offered the option to have more than one supervised session during their initial training on self-injecting in order to improve their confidence and competence, followed by regular periodic reviews. The treatment experience could be optimised by selecting the most suitable device and drug formulation, if available, which would also improve adherence to treatment. Troubleshooting charts or pathways should be provided and explained to patients (e.g., about side effects, problems with the device, missed doses, when to contact the HCPs, etc.), and patients should be able to contact the HCPs in a timely manner if they have any questions. Patients should be given detailed instructions on how to store their injectable drugs and how to monitor the temperature in refrigerators. Ideally, they should be provided with the appropriate equipment for cold storage (e.g., validated pharmaceutical refrigerators, temperature-monitoring devices, and insulated transport boxes for travelling).

Although this qualitative study was limited to 45 patients, the sample size was deemed to be sufficient to reach theoretical saturation for a qualitative, emerging-grounded-theory type of study. Other researchers [34] have reported that the most common sample sizes in qualitative research were 20 and 30 participants, and that general guidelines for reaching saturation in a qualitative study, irrespective of the methodology, suggest between a minimum of 15 and a maximum of 50 participants. 

As explained by Schrijvers et al. [35], a structured care pathway aims to “*enhance the quality of care across the continuum by improving risk-adjusted patient outcomes, promoting patient safety, increasing patient satisfaction, and optimising the use of resources.*” The integrated care pathways help in communication with patients, giving them an informed, printed summary of their expected plan of care; decrease unwanted variation in practice; improve clinician–patient communication; and improve clinical outcomes and patient satisfaction [36]. A standardised education pathway template for discharging patients on the self-administration of injectable therapy was developed for the Trust, which was adapted from O’Prey [37] and Gibson [38].

The proposed education pathway template and training package for discharging patients on HPT (Appendix A) was presented to and discussed with two specialist nurses and a pharmacist involved with HPT. In general, the feedback received about the training package was positive. One of the nurses made a comment that it is always better to start with a more detailed training approach, which can then be tailored to the patients’ needs based on actual training experience. They also commented that, even though the training package is very generic, in their opinion, it does not require a major change as it is well-structured and informative. The HCPs also said that they believed this would give a consistent approach and structure to the training of patients and provide useful information to the patients who are about to start their injectable therapy at home. Another point of discussion was whether patients infected with blood-borne viruses (BBV), like hepatitis-C and human immunodeficiency virus (HIV), and their carers needed additional training regarding the safe handling of injectables, prevention of needlestick injuries, handling needlestick injuries, and how to safely deal with sharps disposal at home, in order to protect themselves and other people (i.e., family members) living in the same household, or the community waste management handlers [39]. It was also discussed whether carers should be involved in the administration of HPT or whether they should be excluded from the care of patients with BBVs in the homecare setting. Selecting the most appropriate injector device, based on its technology and safety features, should be considered in patients with BBVs starting on HPT, to minimise the risk of needlestick injuries and prevent bloodborne pathogen transmission in the homecare setting. A full example of the educational material developed for IBD patients is shown in Appendix A. Similar booklets were also developed for other medical conditions treated with HPT. An integral part of meeting the demand and challenges for the increased self-care of patients at home, in order to improve the quality of care, a range of different integrated supports and services are required [40]. The analysis of the patient interviews confirmed that similar issues with HPT exist in the UK, compared to the rest of the world. For example, patients reported communication barriers and significant problems with their self-injecting (bruising, pain at the injection site, discomfort, anxiety, and problems with injector devices), which was also reported in the study by Schiff et al. [41] in the USA. Compared with other UK studies [1], this study also explored the patients’ experience and perceptions of training, education, and support with HPT. This is believed to be the largest qualitative study in the UK to date that obtained views from participants on HPT with a diverse range of health conditions and explored their experience with HPT and training. In contrast to the study by Twiddy et al. [1], the focus group methodology was not used for several reasons. Focus groups would be more difficult and complex to organise, recruit, and manage, especially as the study was conducted during the COVID-19 pandemic, and may also be inappropriate for exploring sensitive issues that participants may feel uncomfortable discussing in a group discussion. Interviews, on the other hand, allow a much deeper and personalised approach in exploring patient’s experiences and views and are easier to analyse than focus groups, while the role of the interviewer is less important and induces less bias during an interview [42]. 

## 7. Limitations

The findings and themes identified in this study may not be completely generalisable to patients’ receiving HPT in other parts of the UK or globally. Also, this study is limited to patients under the care of two local NHS Trust hospitals. Other NHS Trusts may have a different approach to homecare, and more structured training, staff, and patient support for the conditions’ population. The study was also limited to adult patients (18 years and over) who were currently in the hospital homecare database for the treatment of MS, IBD, psoriasis, atopic dermatitis, high cholesterol, and HPN. Another limitation of this study is a potential selection and response bias because the participants were self-selected. Their interest and willingness to participate in the study may have been associated with some positive or negative aspects or experiences of homecare therapy. 

Another limitation was that the patients were recruited during the COVID-19 pandemic, and some answers may have been biased by patients’ current concerns (e.g., injecting immunosuppressive therapy while shielding at home). Also, the approachability of HCPs was limited during the pandemic for various reasons (e.g., redeployment to COVID-19 areas within the hospital). Moreover, homecare companies were not able to deliver one-to-one training on injectable therapy to some patients due to a number of national lockdowns.

A further limitation of this study was that patients were not asked to read the interview transcripts and provide further comments, explanations, or feedback [43].

## 8. Conclusions

Even when appearing to succeed in managing these therapies, many patients have concerns and questions that arise in the home and require support. Competence in delivering their therapy often does not bring confidence in managing it. Whilst some centres have progressed further in addressing some of these issues, there is evidence to suggest a pan-NHS approach could improve standards and patient outcomes. When patients are allowed to be treated with HPT, significant medical ‘at home’ decisions are delegated to non-medical individuals. Therefore, patient and/or carer education is a vital component of safety. Poor education and communication can leave patients at risk of therapy-related events or lacking the knowledge and confidence needed to be competent collaborators in their own care, and can affect their perceptions of the service, even when they have positive health outcomes.

Based on these results, the implementation of a patient education pathway was suggested, comprising a standardised training package, aiming to reduce unwanted variation in practice, improve clinician–patient communication, and improve patients’ clinical outcomes, support, and satisfaction with HPT. The question remains as to whether HPT training varies between different patient groups based on their disease, or between different HCP teams providing the training, which further supports the idea of a standardised approach to training, regardless of the disease, or clinical or homecare team involved.

It has been shown that, although the patient participants discussed various strengths and positive aspects and experiences with HPT and the homecare programme, they also identified and reported several concerns and challenges they were facing whilst self-injecting therapy at home. 

Many of these challenges reported by patients could have been addressed and/or prevented during the initial training or continued support throughout their therapy. It is also very concerning that some patients did not receive any face-to-face training about their HPT when they were referred to a homecare provider in the community. 

The main finding here was that patients’ experience with HPT is not always aligned with the expectations and beliefs of HCPs. The patient education pathway (including the training material and the training process) should provide a structured, standardised, and documented training package to all patients (and/or their carers) who are discharged from the hospital to the community or their own home on self-injectable therapy, and should be, preferably, implemented on a national level. This is to decrease unwanted variation in practice, improve clinician–patient communication, and improve clinical outcomes and patient satisfaction.

A consistent patient education programme before discharge would likely optimise adherence, effectiveness, and patient confidence with injectable therapy. 

## Figures and Tables

**Figure 1 pharmacy-11-00133-f001:**
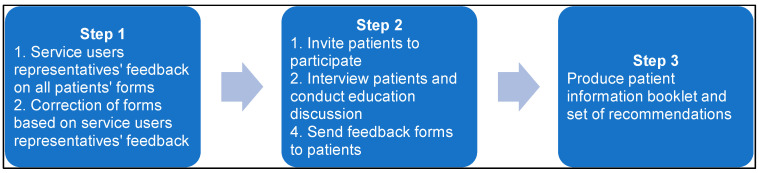
Project steps.

**Table 1 pharmacy-11-00133-t001:** Demographics of interviewed participants (n = 45).

	Frequency	Percentage
Gender—Male	19	42.2%
Gender—Female	26	57.8%
Age group		
18 to 24 years	1	2.2%
25 to 34 years	4	8.9%
35 to 44 years	7	15.6%
45 to 54 years	9	20.0%
55 to 64 years	12	26.7%
65 years or over	12	26.7%
Medication		
Adalimumab	18	40.0%
Glatiramer acetate	6	13.3%
Alirocumab	5	11.1%
Evolocumab	5	11.1%
Ustekinumab	4	8.9%
Peginterferon beta-1a	2	4.4%
Home parenteral nutrition	2	4.4%
Dupilumab	1	2.2%
Interferon beta	1	2.2%
Ixekizumab	1	2.2%
Therapy/Condition		
Inflammatory bowel disease	15	33.3%
(Crohn’s disease)	9	(20.0%)
(Ulcerative colitis)	6	(13.3%)
High cholesterol	10	22.2%
Multiple sclerosis	9	20.0%
Psoriasis	8	17.8%
Intestinal failure	2	4.4%
Atopic dermatitis	1	2.2%

**Table 2 pharmacy-11-00133-t002:** Interview data thematic framework—helpful content for the app or Internet page.

Theme	Number of Participants Who Discussed the Theme	Number of Times the Theme was Mentioned
** *Helpful content for app or Internet page* **	41	69
Question and answers chat or support sections	18	22
Section on side effects	9	9
Video on how to inject, and how it works	7	7
Reminder when to take your injections	2	2
Ability to track and store results	2	2
Understanding the medication and condition	2	2

**Table 3 pharmacy-11-00133-t003:** Interview data thematic framework.

Themes	Number of Participants Who Discussed the Theme	Number of Times the Theme Was Mentioned
**Strengths**	36	107
Positive experience with homecare and initial training	33	47
Feeling supported well, reassurance given	18	22
Feeling confident and positive	10	11
Ability to contact HCPs for advice	8	11
Effectiveness of treatment	6	6
Allowing independence and freedom	4	4
Saving resources	3	3
Simple or easy process	3	3
**Challenges**	42	346
Concerns about injecting the drug	37	146
Concerns about side effects	22	48
Injection site problems	22	51
Problems with injecting technique	11	18
Painful injections	10	18
Use of alcohol wipes	4	5
Problems with injection device	3	4
Preference for method of injection or device	3	3
Anxiety with self-injecting, fear of injections	20	31
Not feeling confident	5	8
Concerns about efficacy	5	8
Dealing with mistakes, troubleshooting	4	5
Being ill and self-injecting	1	1
Missing a dose	1	2
Communication issues	23	53
Inability to contact HCPs for advice	15	28
Reluctance to reach out for help	8	10
Feeling isolated	6	9
Contradictory messages from HCPs	2	4
Poor communication between HCPs	2	2
Perceived lack of support, guidance, and follow-up	22	56
Inadequate training	7	12
Lack of support because of the distance from the hospital	1	2
Lack of support from homecare company	1	1
Challenges with access to information	20	40
Struggling for information	13	25
Older people and issues with using Internet	4	4
Problems with travelling and special storage requirements (fridge)	15	22
Challenges related to COVID-19 pandemic	8	13
Missing reassurance—COVID-19 and weakened immunity	5	9
Face-to-face training missing	1	2
Inability to contact HCPs for advice	1	1
Patients’ scepticism towards treatment options	4	4
Problems with deliveries or supply	4	5
Negative perceptions about doctors, medicine, and pharmaceutical industry	3	5

**Table 4 pharmacy-11-00133-t004:** Demographics of participants who returned both heiQ questionnaires (baseline and follow-up).

	Frequency	Percentage
Gender—Male	13	34.2%
Gender—Female	25	65.8%
Age group		
18 to 24 years	1	2.6%
25 to 34 years	3	7.9%
35 to 44 years	5	13.2%
45 to 54 years	5	13.2%
55 to 64 years	12	31.6%
65 years or over	12	31.6%
Medication		
Adalimumab	15	39.5%
Glatiramer acetate	5	13.2%
Alirocumab	6	15.8%
Evolocumab	5	13.2%
Ustekinumab	3	7.9%
Peginterferon beta-1a	1	2.6%
Dupilumab	1	2.6%
Interferon beta	1	2.6%
Ixekizumab	1	2.6%
Therapy/Condition		
IBD	12	31.6%
CD	7	(18.4%)
UC	5	(13.2%)
High cholesterol	11	28.9%
MS	7	18.4%
Psoriasis	7	18.4%
Atopic dermatitis	1	2.6%

**Table 5 pharmacy-11-00133-t005:** Summary of group effects for eight heiQ domains at baseline and follow-up.

Domain	Baseline Mean	Follow-up Mean	Mean Change	Group-Change Effect Size *	Percent with Positive Reliable Change	Percent with Net Positive Reliable Change
1. Health-directed behavior	3.15	3.20	0.05	**0.08**	18.4	2.6
2. Positive and active engagement in life	3.14	3.20	0.05	**0.10**	5.3	0.0
3. Self-monitoring and insight	3.27	3.26	−0.01	**−0.01**	10.5	−2.6
4. Constructive attitudes and approaches	3.25	3.27	0.01	**0.03**	10.5	2.6
5. Skill and technique acquisition	3.00	2.97	−0.03	**−0.05**	13.2	−7.9
6. Social integration and support	3.11	3.07	−0.04	**−0.08**	10.5	2.6
7. Health services navigation	3.19	3.09	−0.09	**−0.19**	2.6	−10.5
8. Emotional distress	2.34	2.27	−0.07	**−0.11**	13.2	7.9

* A group-change effect size between 0.2 and 0.5 is conventionally said to be ‘small’ while one between 0.5 and 0.8 is conventionally said to be ‘medium’. If 0.8 or greater, the effect size is ‘large’.

## Data Availability

Further information can be share on reasonable request.

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
