# Peer review of "The Experience of Home Parenteral Therapy: A Thematic Analysis of Patient Interviews"

_pharmacy, 2023, doi:10.3390/pharmacy11050133_

Round 1
Reviewer 1 Report
The authors conducted a telephone interview with patients who were self-administering medications in their home settings. Patients provided an assessment before and after the phone interviews as well. The authors anonymized the responses (verbatum) and grouped them into themes. This manuscript is a nice blend of a summary of the authors' findings weaves with direct quotes from the the patients themselves. Overall this is a nice summary of the patient perspective of home injectables and a valuable point that should be considered when providing home care.
As a reviewer outside the UK, there is some background information missing that would help set the stage for an international audience. Currently the introduction jumps right into what current information is available. Missing form the introduction is a description of what HPT is (I gathered that this is self-administration of any injectable medication - this could be IM, IV, or SQ).
Currently the second sentence of the manuscript is about OPAT, which originally lead me to believe this too was going to be a survey of S-OPAT patients, but in reality no patients were receiving antimicrobials. Adding in a generic description of what HPT will help prevent other readers from coming to this conclusion as well.
Additionally, in the methods and design, it should be mentioned here that this was conducted at 2 NHS Trust institutions. Additionally, please provide a general overview of what the current process is, for onboarding patients starting home injectables. Who does the teaching (RN, MD, pharmacist)? Where is the teaching done (inpatient prior to discharge, outpatient clinic appointment, virtually or via telephone)? Is there a standardized check list or pharmacy requirement for education? Knowing this baseline information helps conceptualize when the authors then go on to discuss whether standardization is required.
Table 1 - Please spell out what "HPN" is or define the abbreviations used in footnotes at the bottom of the tables. Additionally, it would be extremely useful to know the method of injection of all these medications. How many are IM? How many are SQ? Without knowing the method of administration - or even the frequency - it's difficult to apply this study/data to other patients self administering. For example, these results would NOT apply to S-OPAT patients.
Page 5 is where what I assume to be the questions that the patients are being asked are introduced, as headers 4.1-4.10. However, the actual questions are never presented anywhere, therefore the flow of the manuscript is interrupted. Additionally 4.8-4.10 aren't even questions, just general topics. I would suggest including a supplement of the exact interview questions the patients were asked.
I do appreciate the actual patient quotes being included in the manuscript. I would overtly state whether the IRB approval included direct quotes being used in manuscripts. However, the quotes can get lengthy. Since the authors already summarize the themes in their own words (the right justified font) and then provide direct patient quotes (left justified font) it can get repetitive. Not every paragraph needs a direct patient quote. I would make sure that only the truly needed, impactful ones are presented. Make sure they are not being repetitive. Additionally, make sure that the left justified quotes are used more than once throughout the manuscript. For example, the quote from P67 "I'm a wimp, I'll admit it" was used on pages 12 and 15.
Page 14, lines 543-552: These seem to be positive of interacting with HCPs, not a negative or area for improvement. I would consider its current placement and whether it is appropriate here.
Lastly, the title is a little too broad and potentially misleading. It reads as if it is all home parenteral therapy - IV, IM , and SQ. But it's only a select group of patients with 3-4 different disease states. Additionally, the routes of administration is missing, but I suspect it is mostly IM or SQ. The title should be sufficiently altered to better align with the number of patients interviewed, the types of diseases states/medications, and routes of administration.
Author Response
Reviwer 1
- As a reviewer outside the UK, there is some background information missing that would help set the stage for an international audience. Currently the introduction jumps right into what current information is available. Missing form the introduction is a description of what HPT is (I gathered that this is self-administration of any injectable medication - this could be IM, IV, or SQ).
Explanation added lines 36-44
- Currently the second sentence of the manuscript is about OPAT, which originally lead me to believe this too was going to be a survey of S-OPAT patients, but in reality no patients were receiving antimicrobials. Adding in a generic description of what HPT will help prevent other readers from coming to this conclusion as well.
Corrected, only two paragraphs are kept as an example, lines 46-51
- Additionally, in the methods and design, it should be mentioned here that this was conducted at 2 NHS Trust institutions. Additionally, please provide a general overview of what the current process is, for onboarding patients starting home injectables. Who does the teaching (RN, MD, pharmacist)? Where is the teaching done (inpatient prior to discharge, outpatient clinic appointment, virtually or via telephone)? Is there a standardized check list or pharmacy requirement for education? Knowing this baseline information helps conceptualize when the authors then go on to discuss whether standardization is required.
Corrected, lines 99-112
- Table 1 - Please spell out what "HPN" is or define the abbreviations used in footnotes at the bottom of the tables. Additionally, it would be extremely useful to know the method of injection of all these medications. How many are IM? How many are SQ? Without knowing the method of administration - or even the frequency - it's difficult to apply this study/data to other patients self administering. For example, these results would NOT apply to S-OPAT patients.
Corrected
- Page 5 is where what I assume to be the questions that the patients are being asked are introduced, as headers 4.1-4.10. However, the actual questions are never presented anywhere, therefore the flow of the manuscript is interrupted. Additionally 4.8-4.10 aren't even questions, just general topics. I would suggest including a supplement of the exact interview questions the patients were asked.
Added as supplementary 1
- I do appreciate the actual patient quotes being included in the manuscript. I would overtly state whether the IRB approval included direct quotes being used in manuscripts. However, the quotes can get lengthy. Since the authors already summarize the themes in their own words (the right justified font) and then provide direct patient quotes (left justified font) it can get repetitive. Not every paragraph needs a direct patient quote. I would make sure that only the truly needed, impactful ones are presented. Make sure they are not being repetitive. Additionally, make sure that the left justified quotes are used more than once throughout the manuscript. For example, the quote from P67 "I'm a wimp, I'll admit it" was used on pages 12 and 15.
Reduced by about 50% to those only that include new information
- Page 14, lines 543-552: These seem to be positive of interacting with HCPs, not a negative or area for improvement. I would consider its current placement and whether it is appropriate here.
Page 14 interaction with health professionals is positive not negative. Whilst this is generally true, the point being made here is that it is confusing and unsettling to patients when health professionals provide conflicting answers.
- Lastly, the title is a little too broad and potentially misleading. It reads as if it is all home parenteral therapy - IV, IM , and SQ. But it's only a select group of patients with 3-4 different disease states. Additionally, the routes of administration is missing, but I suspect it is mostly IM or SQ. The title should be sufficiently altered to better align with the number of patients interviewed, the types of diseases states/medications, and routes of administration.
Corrected lines 1-2
Thank you
Reviewer 2 Report
Lack of adherence to medical treatment causes major clinical problems. Patients understanding of their medication results in providing better therapeutic benefits. Patients‘ education is the base for patient compliance. Medication non-adherence for patients with chronic diseases is very common. Therefore the presented article is very important.
I have some suggestions to the authors:
1. Introduction - there is nothing about the main theme – Home Parenteral Therapy. Could the authors start from the explanation. Not in all European countries self medication of injectable drugs at home is allowed.
I also feel that the analysis of the studies could be moved to the discussion section.
I also wonder if the two patients in TPN should be included in the study. This is very different procedure than administrating drugs and there are separate studies on that subject (i.e. UK study: Clin Nutr ESPEN. 2018 Apr;24:100-108. Patients' experiences with home parenteral nutrition: A grounded theory study. Christina Wong , Beverley Lucas, Diana Wood).
2. Aim – verse 86 :„ the principle aim of this part“ meaning there is another part. Could all aims of the study be mentioned in this section?
3. Methods and design section:
Verse 111 – „second pack 110 containing the draft educational booklet about HPT“ – I understand that it is the one included in the Supplementary? Could you put the reference to make it clear.
4. Patients’ Interviews Qualitative Analysis: This section is a bit difficult to follow. I feel it would be more clear if the authors summarized the patients interview and make a table with the evidential quotes of the patients. In the present form the reader concentrates on each individual patient answer and loses more general view. The quotes should be made more concise – I think the sentences made in bold are enough.
Author Response
Reviwer 2
- Introduction - there is nothing about the main theme – Home Parenteral Therapy. Could the authors start from the explanation. Not in all European countries self medication of injectable drugs at home is allowed.
Corrected lines 36-44 and 99-112
- I also feel that the analysis of the studies could be moved to the discussion section. I also wonder if the two patients in TPN should be included in the study. This is very different procedure than administrating drugs and there are separate studies on that subject (i.e. UK study: Clin Nutr ESPEN. 2018 Apr;24:100-108. Patients' experiences with home parenteral nutrition: A grounded theory study.Christina Wong , Beverley Lucas, Diana Wood).
Thank you for the comments, we appreciate the advice. The study is conducted and completed, as such it will be reporting bias if we remove the two patients and if we did not provide thematic analysis before the discussion.
- Aim – verse 86 :„ the principle aim of this part“ meaning there is another part. Could all aims of the study be mentioned in this section?
Corrected
- Methods and design section:
Verse 111 – „second pack 110 containing the draft educational booklet about HPT“ – I understand that it is the one included in the Supplementary? Could you put the reference to make it clear.
Corrected
- Patients’ Interviews Qualitative Analysis: This section is a bit difficult to follow. I feel it would be more clear if the authors summarized the patients interview and make a table with the evidential quotes of the patients. In the present form the reader concentrates on each individual patient answer and loses more general view. The quotes should be made more concise – I think the sentences made in bold are enough.
Thank you for the comments, we appreciate the advice. We reduced quotations by about 50% to those only that include new information. The paper was written in the traditional qualitative studies format and changing it to a table format will reduce its value.
Thank you
Reviewer 3 Report
In this manuscript, the authors have explored the patients’ experience with home parenteral (injectable) therapy. The target patients were under the care of two local NHS Trust hospitals (UK). I believe that this is a welcome study considering the limited number of studies at the international level.
Major comments:
1. The Introduction section can be completed with other published studies. If you use different similar keywords (e.g. self parenteral therapy), you will find other articles to complete the Introduction section and the main manuscript. In addition, see the template of the journal for references (References should be numbered in order of appearance and indicated by a numeral or numerals in square brackets—e.g., [1] or [2,3], or [4–6].)
2. The method used is appropriate for this type of study. However, I did not find the approval of a research ethics committee before conducting the study, considering that the patients were under the care of two local NHS Trust hospitals. Can the authors provide data in this essential regard?
3. The general impression is that there are too many quotes from the interviews in parallel with the discussion of the main aspects raised by the patients. That is why the manuscript is very long and time-consuming. I suggest organizing them as an appendix (supplementary material) to the article with their citation in the manuscript.
Minor comments:
4. The title is too general.
5. For a better understanding and an overview, I suggest the authors insert a table with the main questions before section 4.1.
6. Regarding the demographic data, it would have been interesting to analyze the patient’s education level (high school, college, etc.) and areas of origin (rural or urban). But these data can no longer be obtained retrospectively.
Author Response
Reviwer 3
- The Introduction section can be completed with other published studies. If you use different similar keywords (e.g. self parenteral therapy), you will find other articles to complete the Introduction section and the main manuscript. In addition, see the template of the journal for references (References should be numbered in order of appearance and indicated by a numeral or numerals in square brackets—e.g., [1] or [2,3], or [4–6].)
Corrected lines 36-44 and 99-112
References are now numbered.
- The method used is appropriate for this type of study. However, I did not find the approval of a research ethics committee before conducting the study, considering that the patients were under the care of two local NHS Trust hospitals. Can the authors provide data in this essential regard?
The trust consists of two big hospitals, the research and ethics department is the same for both. Ethics approvals provided to the editor as confidential documents.
- The general impression is that there are too many quotes from the interviews in parallel with the discussion of the main aspects raised by the patients. That is why the manuscript is very long and time-consuming. I suggest organizing them as an appendix (supplementary material) to the article with their citation in the manuscript.
Thank you for the comments, we appreciate the advice. We reduced quotations by about 50% to those only that include new information. The paper was written in the traditional qualitative studies format and changing it to a table format will reduce its value.
- The title is too general.
Corrected
- For a better understanding and an overview, I suggest the authors insert a table with the main questions before section 4.1.
Added as supplementary 1
- Regarding the demographic data, it would have been interesting to analyse the patient’s education level (high school, college, etc.) and areas of origin (rural or urban). But these data can no longer be obtained retrospectively.
Thank you for the comments, we appreciate the advice.
Thank you
Round 2
Reviewer 1 Report
The authors have addressed the major concerns from both reviewers
Reviewer 3 Report
Thank you for the effort to improve the manuscript and implement all the suggestions.